# 3D Printed Voltammetric Sensor Modified with an Fe(III)-Cluster for the Enzyme-Free Determination of Glucose in Sweat

**DOI:** 10.3390/bios12121156

**Published:** 2022-12-11

**Authors:** Eleni Koukouviti, Alexios K. Plessas, Anastasios Economou, Nikolaos Thomaidis, Giannis S. Papaefstathiou, Christos Kokkinos

**Affiliations:** 1Laboratory of Analytical Chemistry, Department of Chemistry, National and Kapodistrian University of Athens, 15771 Athens, Greece; 2Laboratory of Inorganic Chemistry, Department of Chemistry, National and Kapodistrian University of Athens, 15771 Athens, Greece

**Keywords:** glucose, 3D printing, sensor, nonenzymatic, voltammetry, iron

## Abstract

In this work, a 3D printed sensor modified with a water-stable complex of Fe(III) basic benzoate is presented for the voltammetric detection of glucose (GLU) in acidic epidermal skin conditions. The GLU sensor was produced by the drop-casting of Fe(III)-cluster ethanolic mixture on the surface of a 3D printed electrode fabricated by a carbon black loaded polylactic acid filament. The oxidation of GLU was electrocatalyzed by Fe(III), which was electrochemically generated in-situ by the Fe(III)-cluster precursor. The GLU determination was carried out by differential pulse voltammetry without the interference from common electroactive metabolites presented in sweat (such as urea, uric acid, and lactic acid), offering a limit of detection of 4.3 μmol L^−1^. The exceptional electrochemical performance of [Fe_3_O(PhCO_2_)_6_(H_2_O)_3_]∙PhCO_2_ combined with 3D printing technology forms an innovative and low-cost enzyme-free sensor suitable for noninvasive applications, opening the way for integrated 3D printed wearable biodevices.

## 1. Introduction

Diabetes is one of the most common worldwide diseases. It affects millions of individuals, causing serious damage to the nerves and blood vessels, and ranks among the leading causes of death globally [1]. The periodical checking of blood glucose (GLU) levels throughout the day is of vital significance for diabetic patients, which is typically operated via electrochemical self-testing devices based on blood sampling from the patient’s fingertip. However, this painful blood sampling discourages the patients from frequent measurements during the daylight, while the tests at night-time are practically neglected. On the contrary, noninvasive GLU monitoring is an ideal route toward calm and painless glucose testing, and fortunately modern electrochemical devices have been introduced for GLU monitoring in sweat, as this epidermal biofluid contains GLU at quantities that correlate well with blood [2,3,4,5].

Electrochemical GLU sensors can be split into enzymatic-based and nonenzymatic sensors [6,7,8]. Typical enzymatic GLU biosensors are based on glucose oxidase (GOx) catalyzing the oxidation of glucose to gluconolactone and producing the by-product hydrogen peroxide, which is amperometrically determined by the sensor. The main problems on the construction of enzymatic GLU biosensors are the efficiency of GOx immobilization on the electrode surface, the presence of dissolved oxygen, and the effect of temperature, pH, and ionic strength on the enzyme activity [9,10,11,12]. To overcome these disadvantages, enzyme-free GLU sensors based on metallic particles have been applied as catalysts of GLU electrooxidation, including metals (i.e., Au, Pd and Pt), and metal oxides (i.e., CuO, Cu_2_O, NiO, TiO_2_, Co_3_O_4_, MnO_2,_ Fe_2_O_3_ and Fe_3_O_4_) [13,14,15,16,17,18,19,20,21,22,23,24,25,26], thanks to their high sensitivity, stability and fast response. Nevertheless, these metallic candidates for enzyme-free GLU monitoring usually demonstrate electrocatalytic activity in neutral and basic media and, thus, their applicability in an acidic epidermal skin environment is limited [13,14,15,16,17,18,19,20,21,22,23,24,25,26,27,28,29,30,31,32]. More specifically, regarding the iron-based sensors Fe_3_O_4,_ Fe_2_O_3,_ and FeOOH particles have been used as electrode modifiers for the electrooxidation of GLU in a pH of 7, 7.5, and 13 [27,28,29,30,31,32].

In this work, we have synthesized a water-insoluble Fe(III)-cluster (iron(III) basic benzoate, [Fe_3_O(PhCO_2_)_6_(H_2_O)_3_]∙PhCO_2_) and tested it as an electrode modifier for the enzyme-free, differential pulse voltammetric (DPV) determination of GLU in artificial sweat. The GLU sensor was fabricated using a 3D printed electrode printed from a carbon black loaded polylactic acid (PLA) filament, and the Fe(III)-cluster as ethanolic mixture is drop-casted on the surface of the electrode, followed by its trapping with a Nafion film. The 3D printed sensor was compared with a respective electrode modified with iron oxide that presented poor electrocatalytic ability towards the GLU oxidation in the acidic environment prevailing in epidermal sweat. The cluster of iron (III) basic benzoate exhibited favorable electroanalytical behavior, and once it has been conjugated with 3D printing technology, it leads to an innovative, stable, selective, and sensitive GLU sensor. The core advantages of the presented sensor is its ability to operate in acidic conditions (in contrast to other metallic electrocatalytic GLU sensors), establishing it as suitable for noninvasive applications, while the use of 3D printing overcomes the fabrication disadvantages of manufacturing technologies used in conventional sensors (i.e., screen printed, sputtering), offering low-cost desktop equipment, ease of operation, and fast fabrication and delivery speed with negligible produced waste [33,34,35,36].

## 2. Materials and Methods

### 2.1. Reagents

All reagents were obtained from Sigma-Aldrich (St. Louis, MO, USA). A stock solution of 0.1 mol L^−1^ GLU was prepared in water and left for 24 h at room temperature to allow equilibration of the isomers, and stored at 4 °C. The artificial sweat was composed of 3 mmol L^−1^ NH_4_Cl, 50 μmol L^−1^ MgCl_2_, 0.4 mmol L^−1^ CaCl_2_, 80 mmol L^−1^ NaCl, 8 mmol L^−1^ KCl, 25 μmol L^−1^ uric acid, 22 mmol L^−1^ urea, and 5.5 mmol L^−1^ lactic acid (pH 4) [12]. The phosphate buffer (PB) was prepared by mixing proper quantities of Na_2_HPO_4_ and NaH_2_PO_4_, and the pH value was adjusted to 4 with 1 mol L^−1^ solution of HCl.

### 2.2. Synthesis of [Fe_3_O(PhCO_2_)_6_(H_2_O)_3_]∙PhCO_2_

Following a typical procedure [37], 5.41 g of FeCl_3_∙6H_2_O (20 mmol) were dissolved in 25 mL H_2_O and the solution was filtered to remove insoluble impurities. A second solution was prepared by dissolving 8.65 g of PhCOONa (60 mmol) in 100 mL H_2_O. The two solutions were mixed and stirred for 30 min leading rapidly to the precipitation of a pink-orange powder. The product was isolated by vacuum filtration, it was then washed extensively with H_2_O and was left to dry in air. Yield: 99.2% based on FeCl_3_∙6H_2_O. IR (ATR): 3063 (w), 2164 (w), 1601 (m), 1560 (m), 1493 (w), 1400 (m), 1317 (w), 1177 (w), 1070 (w), 1024 (w), 1001 (w), 941 (w), 839 (w), 818 (w), 712 (m), 685 (m), 675 (m), 660 (w), 631 (m), 482 (m), 411 (w).

### 2.3. Physical Measurements for the Characterization of [Fe_3_O(PhCO_2_)_6_(H_2_O)_3_]∙PhCO_2_

The IR spectrum of [Fe_3_O(PhCO_2_)_6_(H_2_O)_3_]∙PhCO_2_ (Appendix A) was recorded on a Shimadzu FT/IR IRAffinity-1 spectrometer equipped with an ATR unit. Thermogravimetric analysis (TGA) was recorded on a Mettler-Toledo TGA/DSC1 instrument under a N_2_ flow of 50 mL/min from room temperature to 800 °C with a heating rate of 10 °C min^−1^. The powder X-ray diffraction pattern was recorded on a Bruker D8 Advance X-ray diffractometer (CuKa radiation, *λ* = 1.5418 Å). The particle size was calculated using an image taken on a Leica M205 C stereoscope equipped with a Leica DMC5400-20 Megapixel camera (Appendix A). The image was taken from a sample of [Fe_3_O(PhCO_2_)_6_(H_2_O)_3_]∙PhCO_2_, dispersed in water, on a glass slide.

The crystal structure of [Fe_3_O(PhCO_2_)_6_(H_2_O)_3_]∙PhCO_2_ has never been reported by single-crystal x-ray diffraction since the product is formed as a polycrystalline powder, however, similar structures with the same trinuclear moiety, i.e., [Fe_3_O(PhCOO_2_)_6_(H_2_O)_3_]^+^, exist in the literature (Appendix A) [38,39,40]. Furthermore, to the best of our knowledge, the powder diffraction pattern of [Fe_3_O(PhCO_2_)_6_(H_2_O)_3_]∙PhCO_2_ has never been reported. Herein, we manage to elucidate the structure of [Fe_3_O(PhCO_2_)_6_(H_2_O)_3_]∙PhCO_2_ by applying the experimental X-ray powder diffraction data on a model treated by a simulated annealing method and refined with the Rietveld method (Figure 1, Appendix A) using EXPO 1.20.03 [41]. In order to create a suitable model, we examined all known [Fe_3_O(PhCOO_2_)_6_(H_2_O)_3_]^+^ containing structures [38,39,40] from which we eliminated the solvates and replaced the counter anions with a PhCO_2^−^_, using the Avogadro 1.2.0 software [42]. The best results were acquired when we utilized the [Fe_3_O(PhCOO_2_)_6_(H_2_O)_3_]^+^ cation from the crystal structure of [Fe_3_O(PhCOO_2_)_6_(H_2_O)_3_]NO_3_∙3MeCN [38].

The structure of basic iron benzoate (Figure 2) consists of three Fe(III) cations bridged by a μ_3_-oxo bridge. Each pair of Fe(III) is bridged circumferentially by two benzoate anions through their carboxylate groups. Finally, the coordination sphere of each Fe(III) is completed by a terminal H_2_O molecule (whose H atoms could not be modelled), and the total charge is balanced by a benzoate anion in the lattice.

Thermogravimetric analysis (TGA) (Appendix A) reveals that [Fe_3_O(PhCO_2_)_6_(H_2_O)_3_]∙PhCO_2_ loses ~4.29% within the 25–137 °C temperature range, corresponding to the three coordinated H_2_O molecules (their theoretical value is 4.98%). This is followed by a loss of ~10.63% within the 137–257 °C temperature range, corresponding to one benzoate anion (its theoretical value is 11.15%) and degrades immediately after that. The residue above 620 °C is 22.18% which corresponds to Fe_2_O_3_ (theoretical value 22.06%).

### 2.4. Fabrication of the 3D Printed Sensor Modified with Fe(III)-Cluster

The fabrication process of the 3D printed GLU sensor modified with Fe(III)-cluster is illustrated in Figure 3. The 3D printed electrode (3DPE) was designed with Tinkercad software and the printing conditions were set to 60 °C for the platform, 200 °C for the head dispenser, and 60 mm s^−1^ for the printing speed. Flashprint software was used for printing. The filament had a diameter of 1.75 mm and was PLA loaded with carbon black, and was obtained from Proto Pasta. For the construction of the GLU sensor, 10 µL of 6% (*w*/*v*) ethanolic mixture of [Fe_3_O(PhCO_2_)_6_(H_2_O)_3_]∙PhCO_2_, was applied on the cyclic surface of the 3D printer and left for 5 min for its immobilization, followed by curing with an air stream from a gun for 1 extra min. Next, 10 µL of 1% (*w*/*v*) ethanolic solution of Nafion was added on the electrode cyclic surface and left to dry for 5 min. After that, the sensor was treated under an air stream for 1 min for complete drying.

### 2.5. Electrochemical Measurements

The electrochemical measurements were conducted in a 5 mL electrochemical cell in the presence of dissolved oxygen. The portable potentiostat was the EmStat3 (Palm Sens, Houten, The Netherlands) and operated by the PS Trace 4.2 software (Palm Sens, Houten, The Netherlands). The reference electrode was a saturated calomel electrode and the counter electrode was Pt wire. For the DPV measurements, a potential of −1.4 V for 360 s was applied on the 3D printed working electrode (WE), and then a scan (modulation amplitude, 50 mV; increment, 10 mV; pulse width, 75 ms; pulse repeat time, 50 ms) was run on the WE and the DPV response was recorded. The connection of the three electrodes to the portable potentiostat was accomplished using three crocodile clips.

## 3. Results and Discussion

### 3.1. Electrochemical Characterization of the 3D Printed Electrode Modified with Fe_3_O(PhCO_2_)_6_(H_2_O)_3_]∙PhCO_2_

Figure 4 depicts the DPV responses of the 3DPE modified with Fe_3_O(PhCO_2_)_6_(H_2_O)_3_]∙PhCO_2_ and the respective 3DPE modified with iron oxide towards 200 µmol L^−1^ GLU in PB (pH 4). The 3DPE modified with Fe(III)-cluster presented favorable performance offering a well-shaped DPV oxidation peak of GLU, while the respective 3DPE modified with iron oxide exhibited neglected response for GLU oxidation in these acidic conditions. It is has been documented before that Fe_3_O_4,_ Fe_2_O_3,_ and FeOOH based electrodes require neutral or basic conditions in order to form Fe(III), which effectively catalyzed the oxidation of GLU [27,28,29,30,31,32]. The mechanism of electrocatalyzed oxidation of the GLU by the 3DPE modified with Fe_3_O(PhCO_2_)_6_(H_2_O)_3_]∙PhCO_2_ is based on the reduction of Fe(III) in the cluster to Fe(0) on the 3DPE surface, by setting a negative potential at −1.4 V for 360 s. Next, the metallic Fe(0) formed under this cathodic polarization process was oxidized to Fe(III) in the course of the scan potential from −1.4 V to +1.5 V. Finally, the in-situ electrogenerated Fe(III) oxidized GLU [27,28,29,30,31,32]. The whole mechanism is the following:

Mechanism:

(A) Electrogeneration of Fe(III):

Reduction: Fe(III)-Cluster → Fe(0) (polarization of WE at −1.4 V for 360 s).

Oxidation: Fe(0) → Fe(III) (DPV scan from −1.4 V to +1.5 V).

(B) Catalytic oxidation of GLU:

2Fe(III) + Glucose → 2Fe(II) + Gluconolactone + H_2_O.

### 3.2. Effect of Reduction Time, Potential and Fe(III)-Cluster Loading on GLU Determination

The effect of the loading of the Fe_3_O(PhCO_2_)_6_(H_2_O)_3_]∙PhCO_2_ on the 3DPE surface, the reduction potential, and the reduction time of the 3DPE for the in-situ electrogeneration of Fe(III) were examined on the DPV response of 200 µmol L^−1^ GLU in 0.1 mol L^−1^ PB (pH 4) (Figure 5). Four loading levels of the Fe_3_O(PhCO_2_)_6_(H_2_O)_3_]∙PhCO_2_ on the 3DPE in the range 2–8% (*w*/*v*) (as ethanolic mixtures), step of 2%, were studied. As demonstrated in Figure 5, the Fe_3_O(PhCO_2_)_6_(H_2_O)_3_]∙PhCO_2_/3DPE at 6% (*w*/*v*) yielded approximately 1.5 times higher voltammetric peak height of GLU than that of 4% and 2.5 higher than that of 2% (*w*/*v*) loadings, while its sensitivity was statistically comparable with that of 8% (*w*/*v*) loading. Hence, a Fe_3_O(PhCO_2_)_6_(H_2_O)_3_]∙PhCO_2_/3DPE at 6% (*w*/*v*) loading was selected as the optimum, combining the minimal consumption of Fe(III)-cluster with the high DPV response of GLU in acidic conditions.

The effect of the reduction time and corresponding potential of the Fe(III)-cluster on the 3DPE surface was tested in the range 0 s to 480 s and from −1.6 V to 0.0 V, respectively. These parameters affect the quantity of the in-situ electrogenerated Fe(III) and, as a result, the catalytic electrocapability of the 3D printed sensor to GLU determination. As depicted in Figure 6A,B, the GLU oxidation responses increased with respect to the reduction period of time, while a sigmoidal shape is observed that shows a dependence on the reduction potentials. The GLU peak heights were low at more positive potentials, as these potential values were not adequately negative to establish the reduction of the Fe(III)-cluster to metallic Fe(0) on the WE surface. At more negative potentials, the deposition of Fe(0) on 3DPE was favored and the peak currents of GLU increased rapidly up to −1.4 V, where it leveled-off. For the further experiments, a reduction potential of −1.4 V for 360 s was selected, as this presented a satisfactory compromise between high sensitivity and short analysis times.

### 3.3. Analytical Features of 3D Printed GLU Sensor

Figure 7 presents the DPV responses and the calibration plot of GLU on the Fe_3_O(PhCO_2_)_6_(H_2_O)_3_]∙PhCO_2_/3DPE in the concentration range of 25 to 500 μmol L^−1^ in 0.1 mol L^−1^ PB (pH 4). The voltammetric response of the 3D printed sensor to GLU oxidation increased linearly with increasing GLU concentration, with a correlation coefficient of 0.998, and the calibration curve fell within the physiological levels of GLU secreted in human sweat [12]. The limit of detection (LOD) was calculated by the equation LOD = 3 s_y_/a, where s_y_ is the standard deviation of the y-residuals of the calibration plot, and a is the slope of the calibration, which was 4.3 μmol L^−1^. The LOD of GLU achieved with the Fe_3_O(PhCO_2_)_6_(H_2_O)_3_]∙PhCO_2_/3DPE in acidic sweat conditions compares well with those achieved with other iron-based enzyme-free electrodes operated in neutral and basic media [28,29,30,31,32] (Table 1). The within-sensor reproducibility (stated as the % relative standard deviation (RSD) of ten repetitive responses at the 3DPE) was 4.8% for GLU and the between-sensor reproducibility (expressed as the % RSD at six different 3DPEs) was 8.1% (both at the 250 μmol L^−1^ GLU level), revealing high precision of the modified 3DPEs responses.

### 3.4. Interference Study

For enzyme-free GLU sensors, selectivity is a key factor for their noninvasive applications, as the sensors can be subject to interferences by other biomarkers (such as urea, uric acid, and lactic acid) co-existing in sweat that can impact the precision of GLU monitoring. To examine the selectivity of Fe_3_O(PhCO_2_)_6_(H_2_O)_3_]∙PhCO_2_/3DPE in sweat, a concentration of 220 mmol L^−1^ urea, 250 μmol L^−1^ uric acid, and 55 mmol L^−1^ lactic acid was added separately and together in the artificial sweat, and their effect on the DPV oxidation peak of 200 µmol L^−1^ GLU was studied [12,43]. As shown in Figure 8A the sweat biomarkers did not cause any statistically significant effect on the DPV GLU oxidation peak, demonstrating the satisfactory selectivity 3D printed GLU sensor to other common co-existing biomarkers in sweat.

### 3.5. Application to Artificial Sweat

In order to assess the applicability of the enzyme-free method in noninvasive bioanalysis, three artificial sweat samples containing 120 μmol L^−1^ (Figure 8B), 200 and 300 μmol L^−1^ GLU were measured by Fe_3_O(PhCO_2_)_6_(H_2_O)_3_]∙PhCO_2_/3DPEs. The artificial sweat was composed of 3 mmol L^−1^ NH_4_Cl, 50 μmol L^−1^ MgCl_2_, 0.4 mmol L^−1^ CaCl_2_, 80 mmol L^−1^ NaCl, 8 mmol L^−1^ KCl, 25 μmol L^−1^ uric acid, 22 mmol L^−1^ urea, and 5.5 mmol L^−1^ lactic acid [12]. The standard addition method was applied for the determination of GLU in the sweat samples, calculating the respective recovery values. Satisfactory recoveries values for GLU were obtained ranging from 97 to 102%. These results demonstrate the accuracy of the 3D printed sensor modified with Fe(III)-cluster to sensitive and selective monitoring of GLU in sweat conditions.

## 4. Conclusions

In this work, we have developed a new type of 3D printed sensor modified with water-stable Fe_3_O(PhCO_2_)_6_(H_2_O)_3_]∙PhCO_2_ for enzyme-free GLU monitoring in acidic epidermal sweat environment. The Fe(III)-cluster served as a Fe(III) precursor used in the electrocatalytic oxidation of GLU. The 3D printed sensor presented favorable electroanalytical action in the DPV selective determination of GLU, offering satisfactory reproducibility and very low LOD. These features combined with 3D printing technology set the presented sensor as an innovative addition to the arena of electrochemical transducers used for noninvasive bioapplications.

## Figures and Tables

**Figure 1 biosensors-12-01156-f001:**
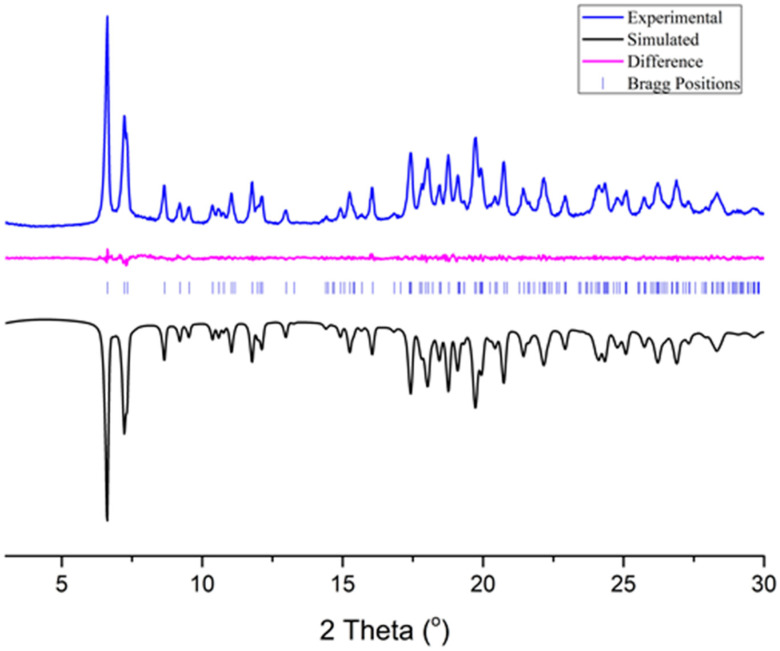
The final plot of the Rietveld refinement, showing the experimental, simulated (reverted) and difference powder diffraction patterns of [Fe_3_O(PhCO_2_)_6_(H_2_O)_3_]∙PhCO_2_. Vertical markers refer to the calculated positions of the Bragg reflections (Appendix A).

**Figure 2 biosensors-12-01156-f002:**
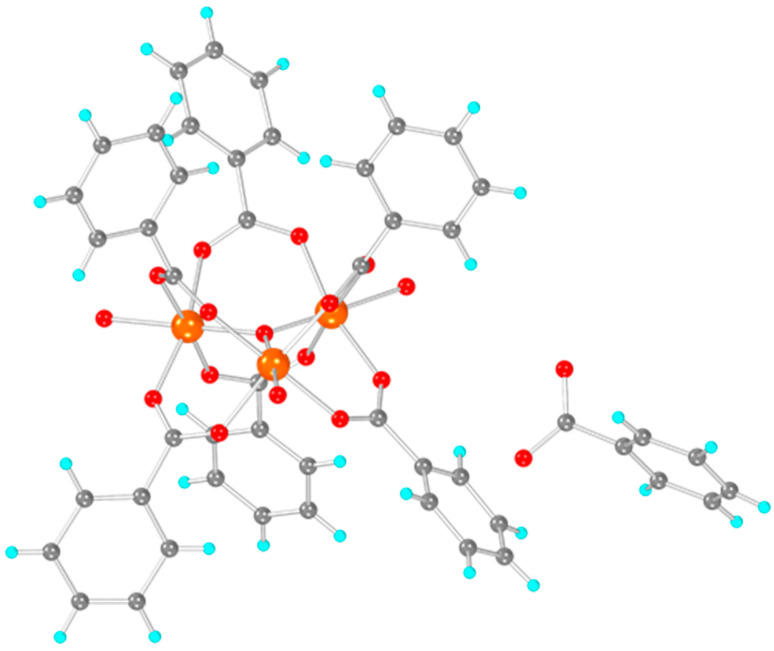
Structure of [Fe_3_O(PhCO_2_)_6_(H_2_O)_3_]∙PhCO_2_ solved by powder X-ray diffraction. Colour code: Fe: orange, C: grey, H: turquoise, O: red.

**Figure 3 biosensors-12-01156-f003:**
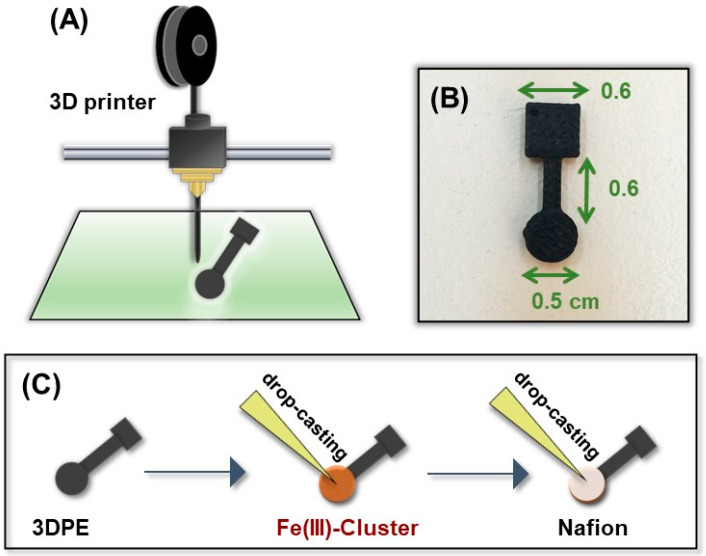
(**A**) Schematic illustration of the 3D printing process of the sensor. (**B**) A photograph of the 3D printed sensor and its dimensions in cm. (**C**) Schematic illustration of the drop-casting procedure for the construction of the GLU 3DPE modified with [Fe_3_O(PhCO_2_)_6_(H_2_O)_3_]∙PhCO_2_.

**Figure 4 biosensors-12-01156-f004:**
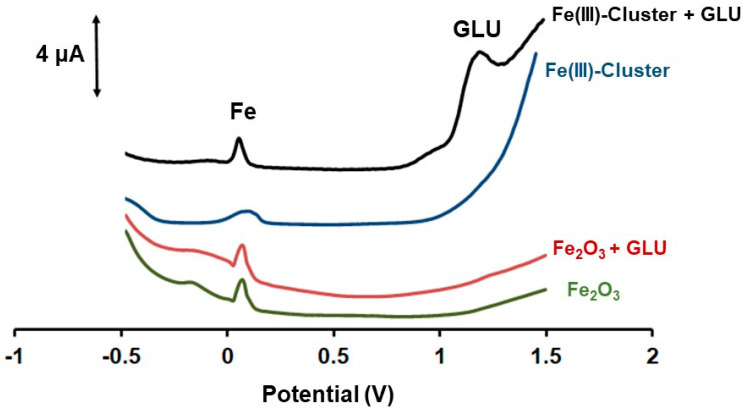
DPV responses of Fe_3_O(PhCO_2_)_6_(H_2_O)_3_]∙PhCO_2_/3DPE and Fe_3_O_2_/3DPE towards 200 µmol L^−1^ GLU in 0.1 mol L^−1^ PB (pH 4). The 3D printed sensor was modified with 6% *w*/*v* Fe_3_O(PhCO_2_)_6_(H_2_O)_3_]∙PhCO_2_ (blue and black lines) and with 6% *w*/*v* Fe_3_O_2_ (red and green lines) (both modifiers as ethanolic mixtures). The reduction potential was −1.4 V for 360 s for Fe(III)-Cluster, while the reduction time was 0 s in cases of iron oxide.

**Figure 5 biosensors-12-01156-f005:**
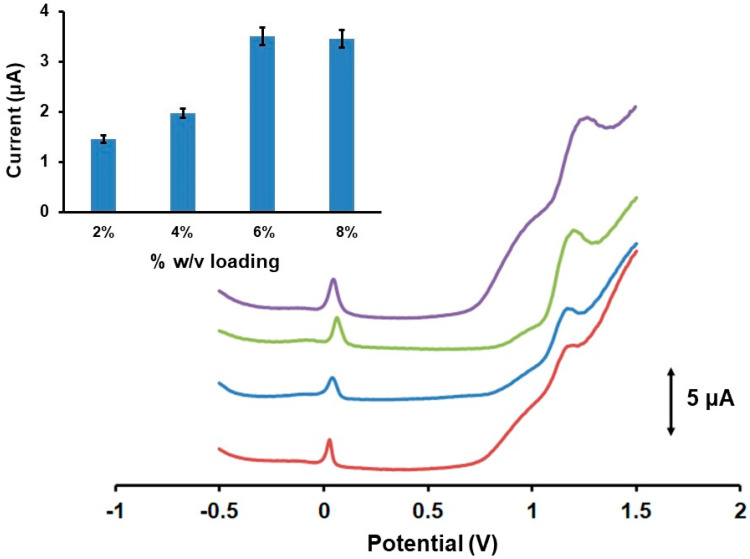
Effect of the concentration of the Fe_3_O(PhCO_2_)_6_(H_2_O)_3_]∙PhCO_2_ at the 3DPE on the DPV peak current values of 200 µmol L^−1^ GLU in 0.1 mol L^−1^ PB (pH 4) and the respective DPV responses (from down to up 2, 4, 6, 8% (*w*/*v*)). Each bar is the mean value ± SD (*n* = 3). Reduction of Fe(III)-cluster at −1.4 V for 360 s.

**Figure 6 biosensors-12-01156-f006:**
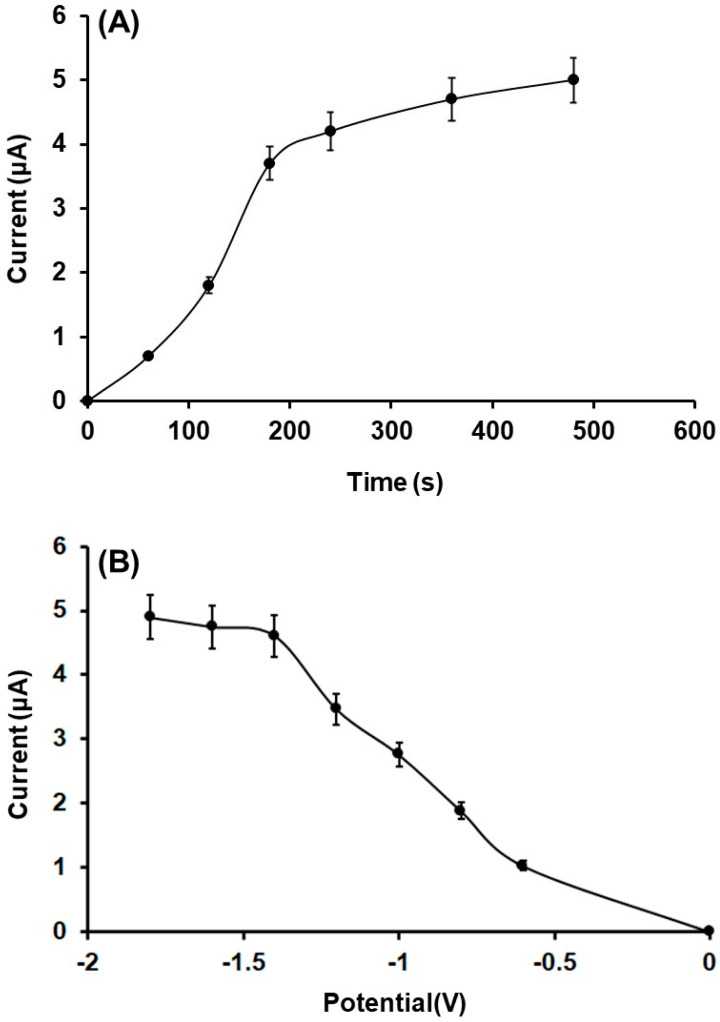
(**A**) Effect of the reduction time on the DPV peak current values of 250 µmol L^−1^ GLU in 0.1 mol L^−1^ PB (pH 4) obtained with Fe_3_O(PhCO_2_)_6_(H_2_O)_3_]∙PhCO_2_/3DPE applying a reduction potential at −1.4 V. (**B**) Effect of the reduction potential on the DPV peak currents of 250 µmol L^−1^ GLU in 0.1 mol L^−1^ PB (pH 4) at Fe_3_O(PhCO_2_)_6_(H_2_O)_3_]∙PhCO_2_/3DPE under 360 s reduction time. Each point is the mean value ± SD (*n* = 3).

**Figure 7 biosensors-12-01156-f007:**
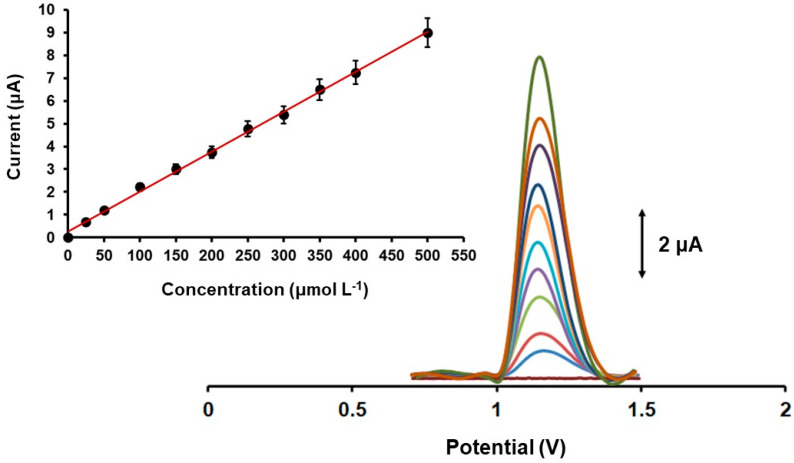
Baseline-corrected DPV responses of GLU concentrations in the range 0–500 μmol L^−1^ (from down to up: 0, 25, 50, 100, 150, 200, 250, 300, 350, 400, 500 μmol L^–1^ GLU) in 0.1 mol L^–1^ PB (pH 4) applying a reduction potential at −1.4 V for 360 s. Each point in the calibration plots is the mean value ± SD (*n* = 3).

**Figure 8 biosensors-12-01156-f008:**
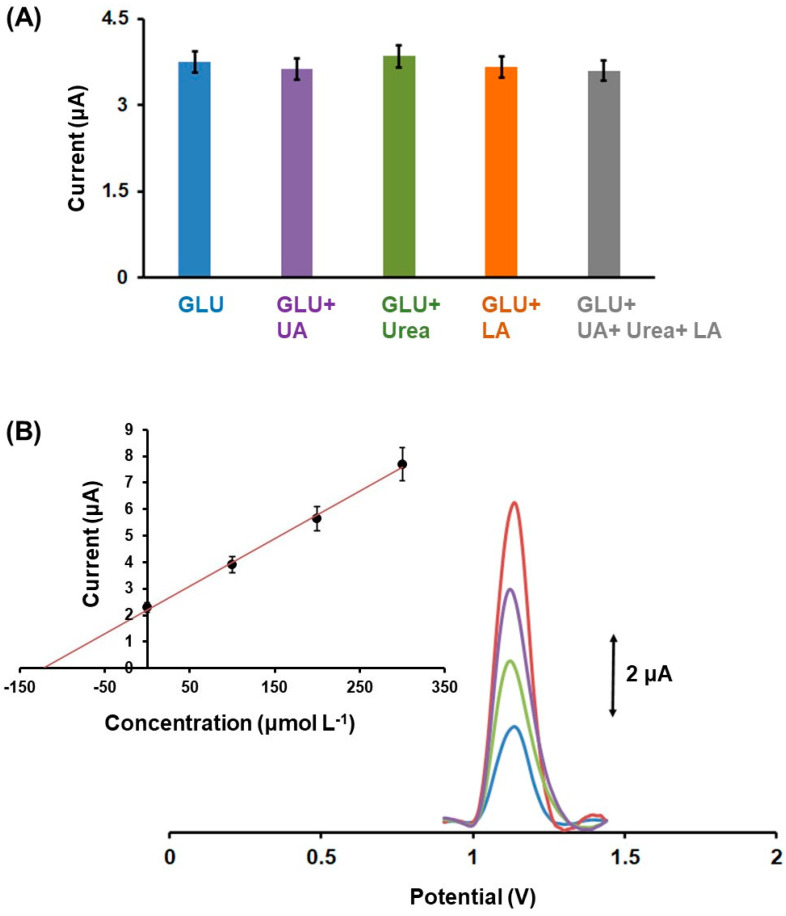
(**A**) Effect of different biomarkers presented in sweat on the DPV peak height of 200 µmol L^−1^ GLU at the Fe_3_O(PhCO_2_)_6_(H_2_O)_3_]∙PhCO_2_/3DPE, where: (blue bar) 200 µmol L^−1^ GLU in artificial sweat (AS); (purple bar) 200 µmol L^−1^ GLU + 250 µmol L^−1^ UA in AS; (green bar) 200 µmol L^−1^ GLU + 220 mmol L^−1^ urea in AS; (orange bar) 200 µmol L^−1^ GLU + 55 mmol L^−1^ lactic acid (LA) in AS; (grey bar) 200 µmol L^−1^ GLU + 250 µmol L^−1^ UA + 220 mmol L^−1^ urea + 55 mmol L^−1^ lactic acid in AS. (**B**) DPV responses and respective plot for the determination of GLU in an artificial sweat sample spiked with 120 µmol L^−1^ GLU. Each point in the bar and in the standard addition plots is the mean value ± SD (*n* = 3).

**Table 1 biosensors-12-01156-t001:** Performance comparison of various iron-based nonenzymatic glucose sensors.

Electrode	Modifier	Operation Media (pH)	LOD (μmol L^−1^)	Ref.
Glassy carbon	Fe_2_O_3_	PB (pH 7.5)	0.6	[28]
Glassy carbon	Fe_2_O_3_	PB (pH 7.5)	6.0	[29]
Glassy carbon	Fe_3_O_4_	PB (pH 7.0)	15.0	[30]
Iron foil	Fe_3_O_4_	NaOH (pH 13.0)	0.1	[31]
Glassy carbon	FeOOH	PB (pH 7.4)	7.8	[32]
3D-CB/PLA	Fe_3_O(PhCO_2_)_6_(H_2_O)_3_]∙PhCO_2_	PB (pH 4.0)	4.3	This work

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
