# Peer review of "3D Printed Voltammetric Sensor Modified with an Fe(III)-Cluster for the Enzyme-Free Determination of Glucose in Sweat"

_biosensors, 2022, doi:10.3390/bios12121156_

Round 1
Reviewer 1 Report
The paper by Koukouviti et al describes the 3d printnig of working electrodes using a conductive PLA filament.
The motivation and overall results are well presented, but some changes must be implemented before further consideration.
- I find missing a summary table of the electroanalytical parameters obtained.
- Morphological characterization of the prepared electrode + modified electrode (SEM, AFM) must be included, if the novelty of the work is based on the preparation of the electrode.
- Comparison of sensitivity and performance regarding similar systems (screen printed electrodes, composite electrodes) is missing.
- How were the dimensions and shape of the printed electrode selected? What about the electroactive area?
- Display / Layout of figures 7 and 8 can be improved, by properly placing the inset in a top corner of the figure (highlighting either the regression graph or the signal graph)
- Why is nafion selected as electrode modifier?
Author Response
Please see our responses in the attached file

Reviewer 2 Report
The manuscript reports on the use of an iron (III) cluster-modified 3D printed electrode for the detection of glucose from acid pH. From an application perspective, artificial sweat spiked with interference, and glucose was used and glucose could be detected from the sweat sample. Although there is a sufficient novelty in terms of the active material, the use of 3D-printed electrodes has already been studied. There are also clarifications to be made both in the materials and also the electrochemistry front. One of my primary concerns is the oxidation of glucose at a high potential of 1.2 V in an acid medium. There are reports where low potential oxidation of glucose has been achieved (ca 0.3–0.5 V). This work needs to be compared in terms of performance with those reports.
Please find my concerns listed:
1. Although the abstract presents a description of the work done, it needs improvement in terms of presenting a background of the work.
2. Also, the first sentence and the second sentence can be combined to give a sentence that encompasses the entire paper with much more clarity.
3. Why have the authors carried out this work in acidic sweat? The pH of sweat is 6.3, so do the authors want to imply that this system is favorable for testing in people with Diabetes also where the pH of sweat is lower?
4. Please make sure that a quantitative value is given for LOD in the abstract for clarity of understanding.
5. “The periodical checking of blood glucose (GLU) levels throughout the day is”- Perhaps “real time” would be a better usage here?
6. However, this painful blood sampling discourages the patients from frequent measurements during the daylight, while the tests at night-time are practically neglected.”- Could the authors please clarify the meaning of this statement? Why is importance given to “day light” and “night time” in this sentence. The pain factor discourages patients from measurements all throughout the day.
7. Sentence construction: “ be split into enzymatic-based and nonenzymatic”
8. Sentence construction: Typical GLU biosensors are based on glucose oxidase- “on the enzyme” is not required.
9. Sentence construction: including metals (i.e Au, Pd, and Pt)
10. Please state the importance of acidic sweat in the introduction.
11. Please combine the sentences “In this work, we have synthesized a water-insoluble Fe(III)-cluster and tested as 56 electrode modifier for the enzyme-free, differential pulse voltametric (DPV) determination of GLU in sweat. The Fe(III)-cluster is the iron(III) basic benzoate, [Fe3O(PhCO2)6(H2O)3]∙PhCO2” for better meaning.
12. An introduction to what is the significance of the use of clusters and why iron cluster was specifically used is required for better understanding of the significance of the work. This introduction can include the properties of the cluster especially the catalytic properties of the cluster would be ideal. Also, out of the many iron clusters that are possible why was iron(III) basic benzoate chose for this application?
13. Please elaborate: IR (ATR): 3063 (w), 2164 (w), 1601 (m), 1560 (m), 1493 (w), 1400 (m), 1317 (w), 1177 (w), 1070 89 (w), 1024 (w), 1001 (w), 941 (w), 839 (w), 818 (w), 712 (m), 685 (m), 675 (m), 660 (w), 631 90 (m), 482 (m), 411 (w).
14. A point to observe is that some of the results like X-ray diffractograms and also TGA have been mentioned before the Results and Discussion. Please make sure that all results are discussed under results and discussion. Also, what is the morphology of the clusters that are used here. I was also not able to find any discussion about the microstructure of the 3D printed electrode. Please give appropriate SEM/TEM images of the clusters, 3D printed electrode and modified electrode to prove this point.
15. Please also mark all the peaks in the X-ray diffractograms. How were the authors able to confirm that the Fe(III) is the oxidation state that is present? Was an XPS done for this?
16. For the DPV measurements, a potential of -1.4V for 360s was applied on the 3D printed working electrode- Could the authors please clarify this statement. The authors have studied the effect of reduction time as part of figure 6. Was the reduction potential applied separately?
17. In figure 4, the oxidation of glucose happens at a high potential of about 1.2 V. The reports of glucose oxidation by iron oxides is observed at comparatively lower potentials. In such an event what are the merits of using this composite for glucose oxidation other than acid pH-based application? In addition there are minor peaks at ca 0.9 V, in both figures 4 and 5. Does this indicate the presence of another phase or incomplete oxidation of glucose by the cluster?
18. Although it is evident from figure 5 that the current demonstrated by 8% loading is higher than 6%, the authors have used 6% loading. Could the authors please explain why such a choice was made? Is the higher potential of glucose oxidation in the case of 8% a reason for this? Please make this justification clearer.
19. Please give proper X-axis values for all the DPVs in the manuscript for ease in comparison.
20. The caption of figure 6 is not clear
21. The authors have given both DP voltammograms and DPV non-uniformly. Please make this usage uniform.
22. The DPVs in figure 5, which are for 200 µM GLU. However, this is not the same when this is plotted in Figure 7. This may be because of difference in the potential range that was adopted. Since the current axis is not mentioned, this cannot be verified.
23. The authors state that they have added 10-fold mass excess of interferents. Please state the mass of the interferents used and also compare that with the mass of glucose used.
24. In the case of the artificial sweat sample what was the pH? Was it 4?
25. In the conclusion part the authors state that the cluster is water stable. How was this tested? Please explain the reason for the water insolubility of the cluster.
26. A table elucidating a comparison of the performance of this sensor with other glucose sensors operating in acid pH should be made.
Hence, I recommend that this manuscript should be revised extensively before a decision is made on whether it can be accepted for publication in Biosensors.
Author Response

(The authors gave the same response as above.)

Reviewer 3 Report
This paper presents a low-cost 3D printed Glucose measuring sensor for noninvasive applications. The reviewer recommends the publication of the manuscript if the following issues are addressed constructively.
1) It is recommended to improve the image in Figure 3c. Maybe typing drop-casting above the image will properly guide the potential readers. At this point, the yellow sharp triangle is hardly understandable.
2) A picture of the actual device that is invasively attached to the body will be helpful.
Author Response

(The authors gave the same response as above.)

Round 2
Reviewer 1 Report
Comments and questions have been properly addressed. The paper is now suitable for publication.
Author Response
We would like to thank the Reviewer for his/her positive evaluation.
Reviewer 2 Report
The authors have not answered many of the questions satisfactorily. My concerns still stand unanswered. Some of them are:
1) The authors state the pH of artificial sweat is 4 and have addressed the use of this sensor in real sweat whose pH they say ranges from 4 to 7.5. In this case according to the argument of the authors, the oxidation peak of glucose depends on pH in addition to the metallic catalyst. This aspect has not been investigated.
2) The nighttime testing of glucose is neglected, which is why real-time testing of glucose is important.
3) The introduction and significance of clusters are not added.
4) The morphology of clusters has not been studied, however, the authors state that the crystal size ranges from 50-200 microns. How was this estimation done?
5) It is understood that the Fe(III) cluster has already been synthesized. In this case, the authors have reproduced the same synthetic techniques and the peaks have been marked. The proof for this reproduction could be seen from the comparison of the peaks and their assignation between the current manuscript and the earlier paper. This has not been done
6) The DPs in figure 5 and figure 7 have mismatches with respect to glucose oxidation. This has not been explained.
7) As part of question 19 in my review, I pointed out that the authors give a justification as to why 6% is better than 8% composition. This justification has not been given.
8) The X-axis values of DPVs are not proper. Some electrochemical characterization figures have been given current values whereas scales have been given for the others. Please explain the reason for this non-uniformity.
9) DP voltammograms and DPV has been used non-uniformly. This error has not been rectified.
10) 10-fold higher interferents have been used and their molarities appear to be different. For example, if 5 mM of glucose is used 50 mM of interferent should be used, which is not the case. An alternative would be to give the weight of the interferents added to the same volume of liquid. This has also not been done.
These are some of the concerns that still exist in the manuscript and hence I recommend that the manuscript cannot be accepted in its present form.
Round 3
Reviewer 2 Report
The authors have answered some questions. However, there are still places where improvements are necessary.
1) I still can find many places where DPV has not been abbreviated.
2) The authors state "Nevertheless, we need to emphasize that the particle size of the Fe(III)-cluster is not important in this work.". However please go through the publication "https://doi.org/10.1038/372346a0" for more information on this aspect. Drawing conclusions from this report, the oxidation of glucose will be influenced by cluster size. Hence, the particle size of the clusters is essential. Hence, an accurate determination of the size of the cluster that you obtained must be made.
3) The peak assignations have not been made. For example, the peak at approximately 7 degrees 2 theta value is because of what plane? Plane assignations are important.
4) It is understood that figure 7 is baseline corrected. Was baseline correction done for the other figures? In addition, the y axis was given for figure 6 and the insets of Figures 5 and 7. But not for the other ones. Please represent all figures uniformly.
Even though it is know that scale bars are acceptable. For example in "Bioelectronics 132 (2019) 136–142" fthe position and intensity of peak R1 in figure 3a is the same as the peak in figure 4a. However in the current manuscript the intensity of the peaks in figure 5 and figure 7 do not match, which is why it is important to give y-axis. This will provide uniformity to representation.
5) Also for clarity, please explain the reasons for taking the particular concentration of interference? Was this taken from some reference?
The manuscript can be accepted if the questions are answered.
